# Validation of a Numerical Bending Model for Sandwich Beams with Textile-Reinforced Cement Faces by Means of Digital Image Correlation

**Jolien Vervloet [1],\*, Tine Tysmans [1], Michael El Kadi [1], Matthias De Munck [1]**, **Panagiotis Kapsalis [1], Petra Van Itterbeeck [2], Jan Wastiels [1] and Danny Van Hemelrijck [1]**

[1] Department Mechanics of Materials and Constructions, Vrije Universiteit Brussel (VUB), Pleinlaan 2, 1050 Brussels, Belgium; tine.tysmans@vub.be (T.T.); michael.el.kadi@vub.be (M.E.K.); matthias.de.munck@vub.be (M.D.M.); panagiotis.kapsalis@vub.be (P.K.); jan.wastiels@vub.be (J.W.); danny.van.hemelrijck@vub.be (D.V.H.)

[2] Department of Structures, Belgian Building Research Institute (BBRI), Avenue P. Holoffe 21, 1342 Limelette, Belgium; petra.van.itterbeeck@bbri.be

\* Correspondence: jolien.vervloet@vub.be; Tel.: +32-(0)2-629-29-24

**Abstract:** Sandwich panels with textile-reinforced cement (TRC) faces merge both structural and insulating performance into one lightweight construction element. To design with sandwich panels, predictive numerical models need to be thoroughly validated, in order to use them with high confidence and reliability. Numerical bending models established in literature have been validated by means of local displacement measurements, but are missing a full surface strain validation. Therefore, four-point bending tests monitored by a digital image correlation system were compared with a numerical bending model, leading to a thorough validation of that numerical model. Monitoring with a digital image correlation (DIC) system gave a highly detailed image of behaviour during bending and the strains in the different materials of the sandwich panel. The measured strains validated the numerical model predictions of, amongst others, the multiple cracking of the TRC tensile face and the shear deformation of the core.

**Keywords:** finite element model; real scale bending experiments; shear; structural insulating sandwich panel

## 1. Introduction

Structural insulating sandwich panels combine a lightweight insulating core with two thin stiff faces, hence they can improve the energy efficiency of the building and provide the necessary structural capacity. Due to this dual function, these panels are gaining more interest from the building industry, as they are very suitable for nearly zero-energy buildings and contribute to reach the energy efficiency objectives of the European Union.

Nowadays, pre-cast concrete sandwich panels are frequently used for walls in residential and commercial buildings, since their energy efficiency and structural capacity are well-known [1–4]. The weight of these concrete sandwich panels can be drastically reduced by replacing the steel-reinforced concrete faces by textile-reinforced cement (TRC) faces. Due to the use of textiles instead of steel, the thick concrete cover (needed for durability reasons in case of steel) can be avoided. This reduces the thickness of the faces, and therefore the weight as well.

The research groups of Hegger et al., Colombo et al., and Cuypers et al. investigated the behaviour of sandwich panels with TRC faces by bending experiments [5–7]. Hegger et al. also added connectors between the two faces to enhance the composite action of the panel [8]. The behaviour of TRC sandwich

panels in compression, as well as their durability, has been recently explored by Tysmans et al. [9–11]. These studies represent the first step towards the application of TRC sandwich panels in residential, public and industrial buildings, as cladding or wall panels [12,13].

In order to accurately and safely design TRC sandwich panels for their application, the prediction of their behaviour under different loading conditions is indispensable. A few analytical models have been already established in [14–16], while numerical models can be found in [17–19]. The established numerical models were validated by experiments measuring the force-displacement behaviour or local strains of the sandwich panels. Accurate full-field strains of the bending behaviour of TRC sandwich panels to validate the existing models are, however, still missing in the current state of the art. Therefore, four-point bending tests monitored by digital image correlation (DIC) were performed in the scope of this paper, and were compared to results of a numerical model.

This paper shows a thorough validation of numerical bending models of TRC sandwich beams, available in literature [6,19], by full-field DIC results on four-point bending experiments. While in previous literature, the validation of the numerical model has been limited by local displacement measurements, this paper shows a detailed comparison of the strains in the faces and the core. The full-field analysis of the DIC measurements reveals four stages in the bending behaviour of the sandwich beams, and shows the behaviour of each component material (faces and core) during the experiments. This provides a more in-depth comparison and shows a good agreement between the numerical prediction and experimental results. As a conclusion, it can be stated that the established numerical model was validated and was able to simulate the behaviour in bending of TRC sandwich panels with high confidence.

## 2. Materials and Methods

### 2.1. Textile-Reinforced Cement

The faces of the used sandwich panels were made of TRC plates consisting of a cement matrix cast onto fibre textiles. The cement matrix was a self-compacting ordinary Portland cement (OPC) composed of CEM I 52.5 R cement, fly ash, silica fume, silica flour, superplasticizer, and a water/cement ratio of 0.15. The cement was commercially available as SikaGrout 217 [20], and had a maximum grain size of 1.6 mm and a density of 2000 kg/m$^3$. The compressive strength and compressive E-modulus were 58 MPa and 26 Gpa, respectively. The compressive strength of the cement was experimentally determined by calculating the average of seventeen cubes with dimensions of 150 mm × 150 mm × 150 mm, in accordance with NBN EN 12390-3 [21]. The E-modulus was measured by applying strain gauges on three cylindrical specimens (VUB, Brussels, Belgium) with a height of 230 mm and a diameter of 104 mm, which were subjected to compression test according to [22].

The textile reinforcement used for the TRC faces was a combination of three-dimensional (3D) and two-dimensional (2D) textiles. The 3D textile was a spacer textile, composed of two layers of 2D textiles kept at a distance of 11 mm by polyester PET fibres. The 3D textile was combined with two 2D textiles, one placed on the top and one on the bottom of the 3D textile, to increase the fibre volume a fraction above the critical one (0.73%). The critical fibre volume fraction has to be exceeded in order to create the strain hardening behaviour of the TRC. The critical fibre volume fraction was calculated by the matrix tensile stress $\sigma_{mu}$, the E-modulus of the matrix $E_m$, the fibre tensile failure stress $\sigma_{fu}$, and the E-modulus of the fibres $E_f$:

$$V_f > \frac{\sigma_{mu}}{-\frac{E_f}{E_m}\sigma_{mu} + \sigma_{mu} + \sigma_{fu}} \tag{1}$$

Both textiles consisted of alkali-resistant (AR) glass fibres placed in an orthogonal grid structure, and are presented in Figure 1. With a thickness of the faces at 22 mm, the total fibre volume fraction was 2.98%, and the effective fibre volume fraction in the loading direction was 1.49%. The specifications of both textiles are given in Table 1.

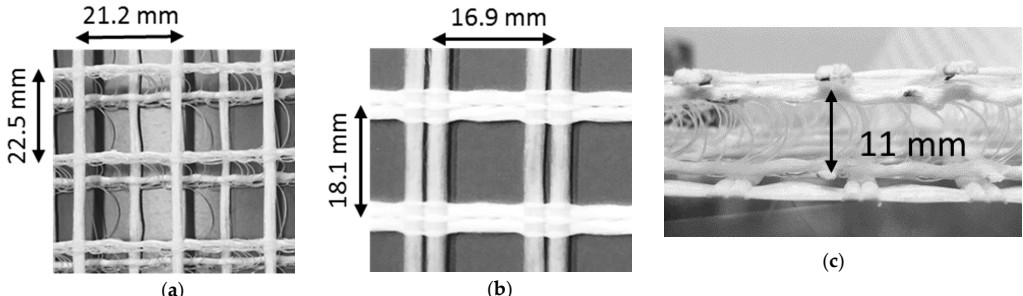

**Figure 1.** Textiles used in textile-reinforced cement (TRC): (**a**) three-dimensional (3D) textile, (**b**) two-dimensional (2D) textile, and (**c**) a combination of 2D and 3D textiles.

**Table 1.** Specifications of both 2D and 3D textiles.

|  | Fibre Material | Density (g/m$^2$) | Spacer Distance (mm) |
|---|---|---|---|
| 3D textile [23] | AR-glass | 536 | 11 |
| 2D textile [24] | AR-glass | 568 | - |

The textile-reinforced cement faces, made for mechanical characterization, were cast in wooden moulds with dimensions as follows: height = 450 mm, width = 500 mm, and thickness = 22 mm. Before placing the 2D and 3D textiles, a layer of 5 mm cement was cast in the moulds. When the textiles were placed, the mould was filled with cement until a thickness of 22 mm was reached. Due to the self-compacting nature and small grain size of the cement, it could easily flow through the textiles and fill the mould. The moulds were covered with plastic foil, and the textile reinforcement cement plates were left to harden for 28 days.

### 2.1.1. Tensile Behaviour of Textile-Reinforced Cement

The tensile behaviour of TRC faces was investigated in detail in [25]. A brief description is given hereafter. The TRC faces of the sandwich beams were tested by a tensile test based on the recommendation of RILEM TC 232-TDT [26]. A schematic presentation of the test is given in Figure 2a. The dimensions of the specimens were as follows: length = 450 mm, width = 59 mm, and thickness = 22 mm. A total of 15 specimens were tested in tension, with a rate of 1 mm/min. The obtained stress–strain behaviour is presented in Figure 2b, which clearly shows three stages (indicated by I, II and III). In the first stage, the matrix and textiles showed composite action resulting in an E-modulus of 10.7 GPa. After reaching the matrix cracking stress of 2.28 MPa the second stage of multiple cracking occurred until crack saturation was reached (at a strain of 0.0033), and resulted in an E-modulus of 0.34 MPa. The third stage—post-cracking—was mainly determined by the textiles, and resulted in an E-modulus of 0.75 GPa and an ultimate stress of 7.43 MPa. The previously mentioned properties were average values of 15 specimens, and were used to establish the average tri-linear tensile stress–strain curve shown in Figure 2b.

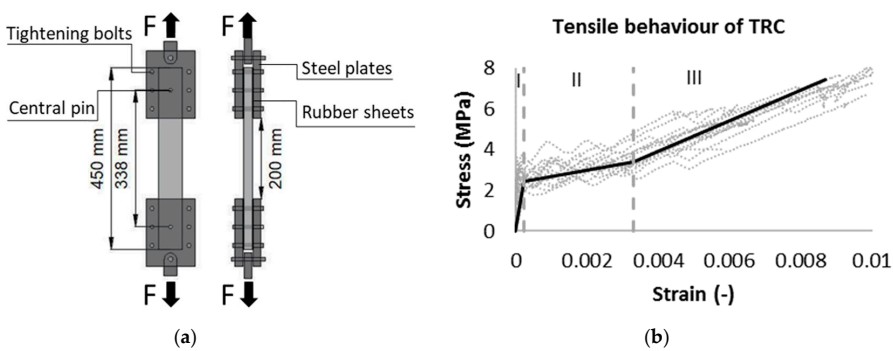

**Figure 2.** (**a**) Tensile test set-up, and (**b**) tensile behaviour of TRC faces.

## 2.2. Extruded Polystyrene Foam

The thermal insulation used for the sandwich beams was extruded polystyrene foam (XPS), in the form of rigid plates with a density of 33.5 kg/m³, experimentally determined from six specimens. The top and bottom surfaces of the rigid insulation plates were imprinted with a rhombus pattern to provide mechanical interlocking and a better stress transfer between the TRC faces and the core. The finishing of the surfaces is shown in Figure 3a. The thickness of the foam blocks was 160 mm.

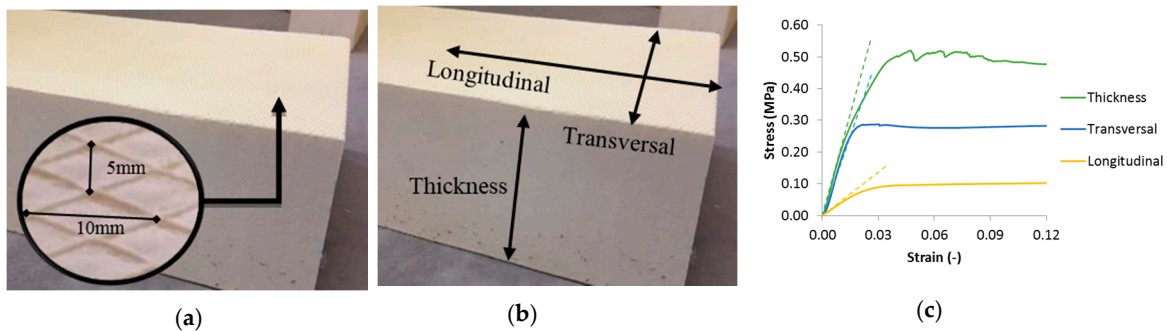

**Figure 3.** (**a**) Rhombus pattern on the faces of the rigid insulating extruded polystyrene foam (XPS) plates, (**b**) directions of the foam, and (**c**) compression test results in the different directions of the XPS foam.

2.2.1. Compressive Behaviour of XPS

Due to the extrusion production technique, the foam behaved differently in all three directions. Four compression tests on XPS cubes in every direction of 160 mm × 160 mm × 160 mm were performed in accordance with ASTM C165 [27], in order to determine the E-modulus and ultimate compressive strength of the foam. The best performance, in terms of stiffness and strength, was found in the thickness direction (see Figure 3c). The ultimate strength for the longitudinal, transversal, and thickness direction equalled 0.09MPa ($\sigma_{cl}$), 0.29 Pa ($\sigma_{ctr}$), and 0.52 MPa ($\sigma_{cth}$), respectively. The E-modulus was equal to 3.61 MPa ($E_{cl}$), 17.04 MPa ($E_{cp}$), and 20.6 MPa ($E_{ct}$) for the longitudinal, transversal, and thickness directions, respectively.

The shear strength and modulus of the XPS foam were determined by bending tests on four sandwich beams, with a span of 1 m and a width of 400 mm each, as described by the standard NBN EN 14,509 (2013) [28] (see Figure 4). This led to a shear modulus ($G_c$) of 4.7 MPa, and an ultimate shear strength ($\tau_c$) of 0.24 MPa.

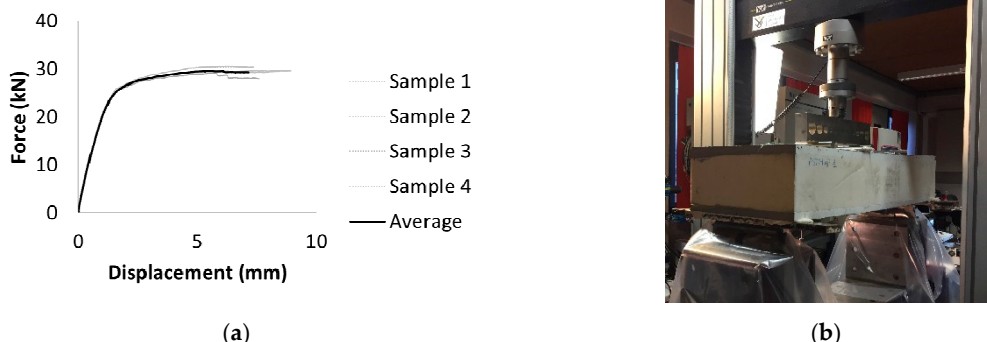

**Figure 4.** (**a**) Force-displacement curves of bending tests performed to determine the shear strength of the core, and (**b**) the set-up of the bending test.

## 2.3. Production of Sandwich Beams

The construction of the sandwich beams was done in multiple phases. First, the XPS foam was placed into the mould so that the transversal direction of the foam (see Figure 3b) was aligned with the

span of the beam. A thin cement layer of 5 mm was cast onto the XPS foam block, on which the 2D and 3D textiles were placed. The advantage of using a 3D textile was that the textile layers were kept directly at the right distance from each other. Afterwards, fresh cement paste was cast on the textile, until the face thickness of 22 mm was achieved. The self-compacting properties of the cement paste made it flow easily through the textile reinforcement and spread over the whole surface. In the next step, the surface was covered to reduce evaporation. After 24 h of hardening, the beam was turned over, and the second face was cast onto the XPS foam in the same way as the first face.

### 2.4. Four-Point Bending Set-Up

The load-deformation behaviour of the sandwich sections was investigated by means of a four-point bending set-up. This set-up allows for an area with a constant moment, where tensile stresses in the lower TRC face dominate. Furthermore, the set-up provoked shear stresses in the core between the supports and loading beams.

Four sandwich beams, with a span of 2.2 m between the supports, were tested in four-point bending with a displacement rate of 10 mm/min. The distance between the applied forces was 0.5 m, while the width and thickness of the beam were 0.4 m and 0.204 m (see Figure 5). The production process of the sandwich panels was explained in Section 2.3. During the test, the specimens were monitored with two DIC systems. DIC is a non-destructive measurement technique that records displacements of the entire observed specimen surface (by means of a speckle pattern), from which strains can be calculated. The displacements are related to a reference image taken at a zero-load step [29]. This measurement technique has proven to provide detailed information on textile and fibre reinforced cement application, as explained in [30] and [31]. The field of view of each system is captured a length of 600 mm along the length of the sandwich beam, starting from the middle of the beam (see Figure 5b).

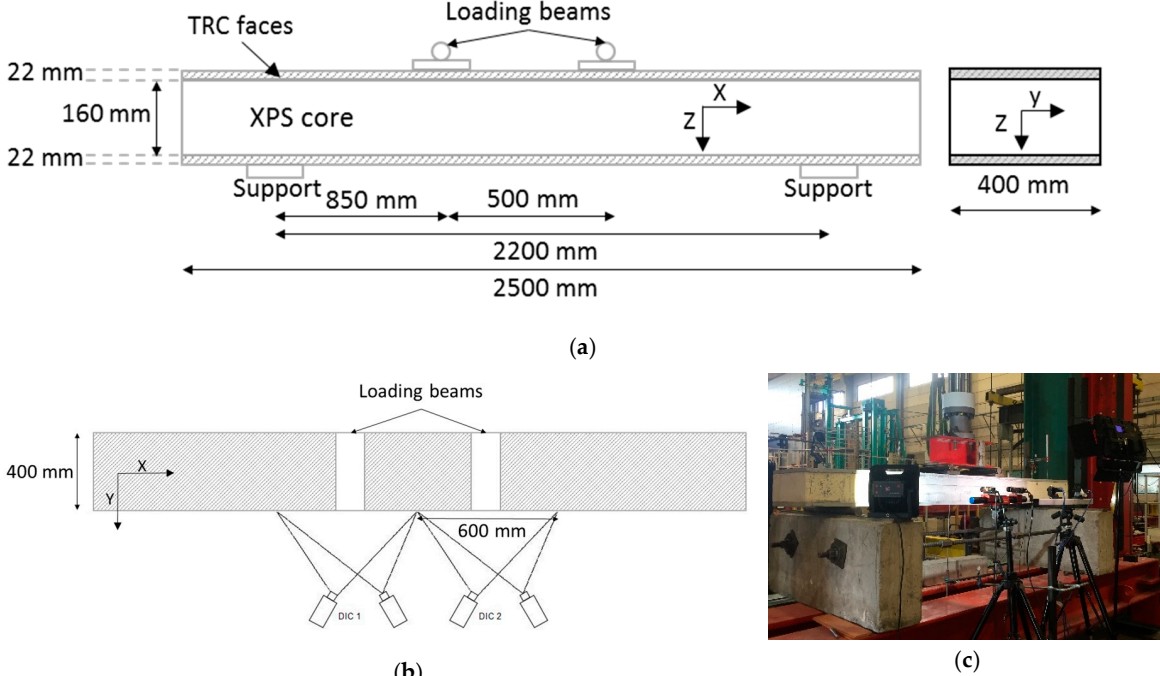

**Figure 5.** (**a**) front view of four-point bending test, (**b**) top view of four-point bending test, and (**c**) picture of test set-up.

## 3. Numerical Model Definition

### 3.1. Material Definition

The numerical modelling was performed in the finite element software ABAQUS/Explicit [32]. Non-linear material behaviour is applied by means of different prescribed material models in the program. The tensile and compressive behaviour of the TRC faces was implemented by combining the elastic and concrete damaged plasticity (CDP) material model. The compressive behaviour of the TRC faces was implemented in the elastic material model. Hence, the linear elastic behaviour was described by the compressive E-modulus of 26 GPa and the Poisson ratio of 0.21 [33] of the cement. For the CDP model, the requested input parameters were the dilation angle ($\psi$ = 36), the potential flow eccentricity ($\varepsilon$ = 0.1), the proportion of the ultimate compressive stress in a biaxial test to the uniaxial compressive stress ($f_{b0}/f_c$ = 1.0), the shape of the deviatoric cross section ($K_c$ = 0.667) and the numerical viscosity parameter ($\mu = 10^{-7}$). The values of these parameters were based on the ones described in [34]. Besides previously mentioned parameters, the non-linear tensile behaviour and ultimate compressive strength of the TRC were implemented in the uniaxial tensile stress-strain input of the CDP model. The compressive strength was limited to 58 MPa, while the complete tensile behaviour of the TRC faces, including the linear elastic part, was inserted. The used characteristic values are shown in Table 2.

**Table 2.** Characteristic values for the average tensile TRC curve.

| Matrix Cracking Stress | Matrix Cracking Strain | End of Multiple Cracking Stress | End of Multiple Cracking Strain | Ultimate Failure Stress | Ultimate Failure Strain |
|---|---|---|---|---|---|
| 2.28 MPa | 0.00022 | 3.38 MPa | 0.0033 | 7.43 MPa | 0.0087 |

The initial linear elastic behaviour of the XPS foam was implemented by the elastic material model, and defined by the E-modulus and the Poisson ratio. The E-modulus, inputted into the numerical model, was based on the previously determined shear characteristics. Linear elastic analytical bending models for sandwich panels, described in [35,36], show that the deflection due to shear (80%) was significantly larger than the deflection due to bending (20%). Therefore, the applied E-modulus was calculated from the shear modulus (see Section 2.2) and the Poisson ratio (0.5) [19] of the XPS core.

The non-linear behaviour of the foam was modelled by the crushable foam–volumetric foam hardening material model. This model took into account the increased deformation of the foam in compression due to buckling of the cell walls, but required an isotropic material [32]. The crushable foam model requires the following parameters: the ratio between the initial yield stress in uniaxial compression and the initial yield stress in hydrostatic compression $\sigma_c^0/p^0_c$ (compression yield stress ratio), as well as the ratio between the hydrostatic tension and the initial yield stress in hydrostatic compression $p_t/p^0_c$ (hydrostatic yield stress ratio). The hydrostatic tension and initial yield stress in hydrostatic compression were set to 0.15 MPa and 0.14 MPa, respectively, as described in [19]. The initial yield stress in uniaxial compression was determined experimentally and set to 0.21 MPa (see Figure 3c). These values led to a compression yield stress ratio of 1.5 and a hydrostatic yield stress ratio of 1.07. The nonlinear behaviour of the foam was implemented through the yield stress and uniaxial plastic strain obtained from the average stress–strain curve of the thickness direction (Figure 3c). This non-linear material model implies the use of a dynamic explicit analysis, which is implemented with a time period of 10 and a mass scaling factor of 0.000001, in order to improve the speed of the analysis.

*3.2. Cross Section and Boundary Conditions Modelling*

A numerical model was established to simulate the four-point bending behaviour of sandwich beams with TRC faces, in order to compare it with experimental results of the same sandwich beams. In this way, more confidence in the numerical model was gained.

Both the faces and the core were defined as solid homogeneous sections in the numerical model and, were discretised by eight-node linear brick elements (C3D8R). The elements size was 35.7 mm × 3.6 mm × 200 mm (w × h × d) for the faces and 35.7 mm × 13.3 mm × 200 mm (w × h × d) for the core. Six elements were stacked over the thickness of the faces, and twelve over the thickness of the core of the sandwich beam. The mesh size was the result of a convergence study on the force-displacement behaviour of the sandwich beam, as well as on the stress and strain output. Multiple elements were necessary throughout the thickness of the faces to evaluate the stress/strain distribution over the thickness. Only one element was assumed over the width of the beam, since the load distribution, and therefore also the beam response, was assumed to be uniform. The mesh is shown in Figure 6.

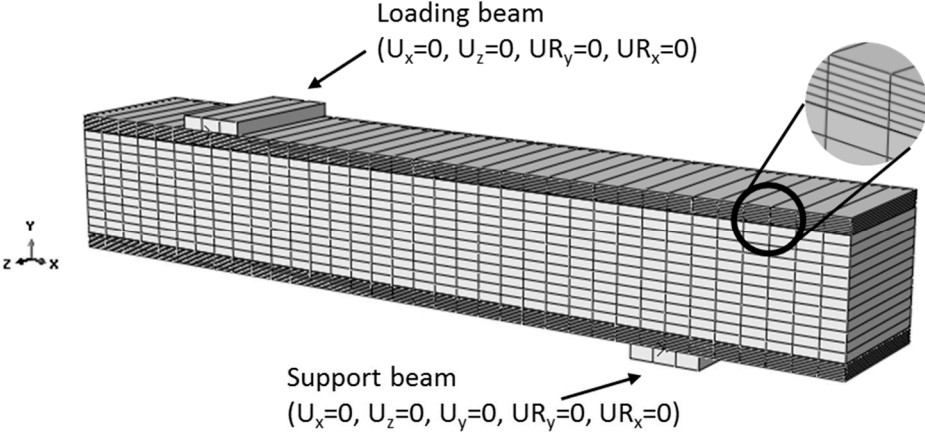

**Figure 6.** Numerical bending setup.

The contact between the rigid bodies (loading beams and supports) and the sandwich panels was established by a frictionless and hard contact interaction. The bond between the core and faces, however, was considered perfect, since no debonding was encountered during the experiments. Hence, the surfaces of the core and faces were modelled by a cohesive surface interaction, without defining damage interaction, defining a perfect bond. The default contact enforcement method was implemented, meaning that the elastic properties of the bond are based on the underlying element stiffness [32].

To simulate the bending behaviour of the sandwich beams, two rigid bodies and symmetry planes were used. In order to limit the number of elements, and therefore also the calculation time, symmetry boundary conditions were used in the *XY* and *YZ* planes. The results, however, can be plotted for the whole beam. One of the rigid bodies represents the support, while the other represents one of the loading beams. The support was restricted in the *X*, *Y* and *Z* directions, and could only rotate around the *Z*-axis. The loading beam was restricted in the *X* and *Z* directions, and could rotate only around the *Z*-axis.

For convergence reasons, the imposed displacement was performed with a smooth amplitude, so that the increments were smaller.

**4. Results and Discussion**

Figure 7a shows the force-displacement graph of a sandwich beam under four-point bending (as described in Section 2.4. Four-Point Bending Set-Up), where the displacement is measured at the tensile face of the beam underneath the loading pins by Linear Variable Differential Transformers (LVDTs). The orange curve shows the prediction by the numerical model, and the blue curve gives

the average of the experimental results. The grey area shows the scatter on the experimental results. During the experimental campaign, four sandwich beams were tested. All sandwich beams failed by shear failure of the core, as shown in Figure 7b.

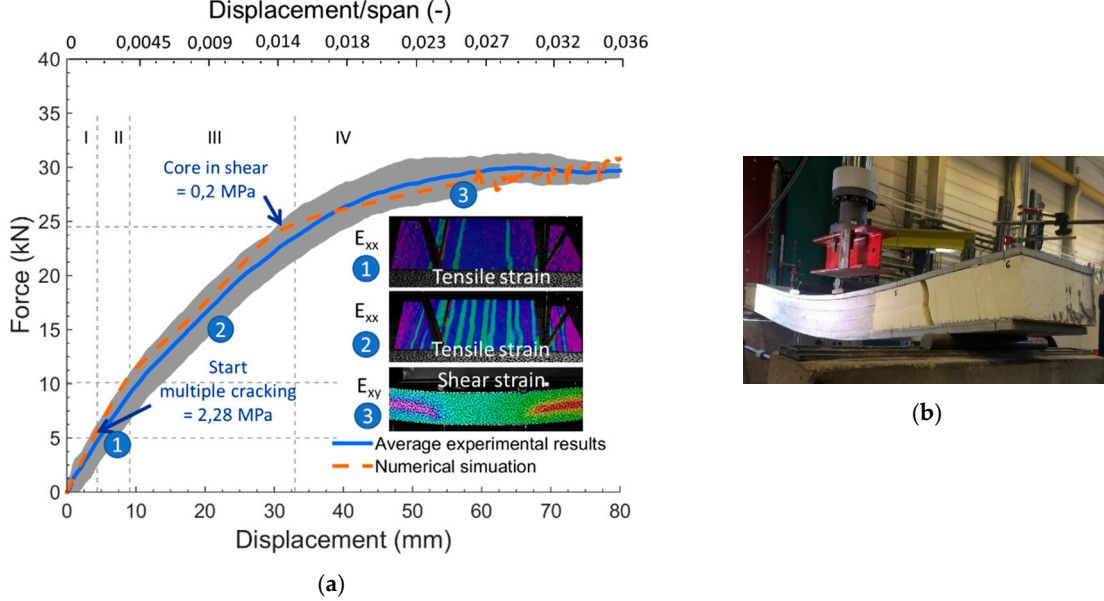

(a)

**Figure 7.** (**a**) force-displacement curve of the four-point bending tests on sandwich beams with TRC faces and the numerical prediction, and (**b**) failure in the shear of the core of a representative sandwich beam.

## 4.1. Numerical Model

The established numerical model revealed multiple stages in the bending behaviour of the TRC sandwich beams, based on the stress and strain development in the different materials of the sandwich beam. Four stages were distinguished, and indicated with I, II, III, and IV in Figure 7a. The first stage showed linear elastic behaviour of the beam. At a load of 5 kN (start of stage II), the matrix cracking stress of 2.28 MPa was reached at the surface of the tensile face, in the area with the constant moment (see Figure 8), which physically corresponds to the initiation of the first crack. The first crack initiation and the development of multiple cracks in the tensile face of the beam are specified for the second stage in Figure 7. Once the matrix cracking stress reaches through the complete thickness of the face (at a load of 10 kN in Figure 7, and illustrated in Figure 9), a clear reduction in stiffness was noticed, leading to the start of the third stage. Starting from a load of 25.5 kN, the core no longer deformed linearly and elastically but plasticly (see Figure 10), which led to another reduction in stiffness and the start of the fourth stage. The part of the plastic shear strain and total shear strain strain are presented in Figures 10 and 11, respectively. The maximum displacement in Figure 7 was a result of the applied maximum displacement of 100 mm during the analysis. Failure of the TRC sandwich beam was obtained when the ultimate shear stress (0.24 MPa) of the core was reached, which happened at a displacement of 91 mm as illustrated in Figure 12.

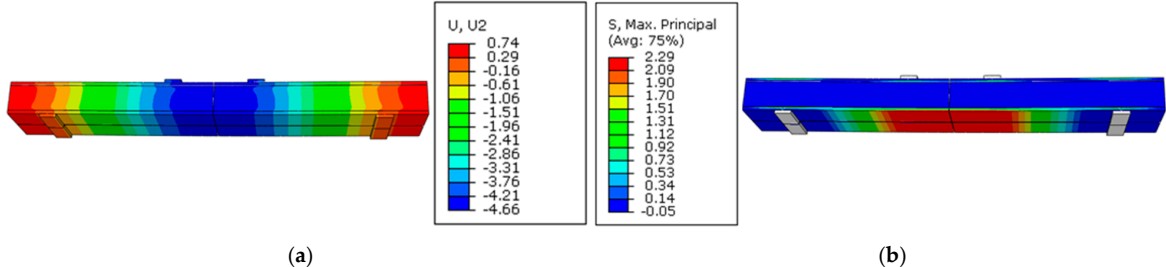

(**a**)　　　　　　　　　　　　　　　　　　　(**b**)

**Figure 8.** Start of the second stage, where the matrix cracking stress is reached at a vertical load of 5 kN. (**a**) Vertical displacement (U, in mm) and (**b**) horizontal stress (S, in MPa) in the tensile TRC face.

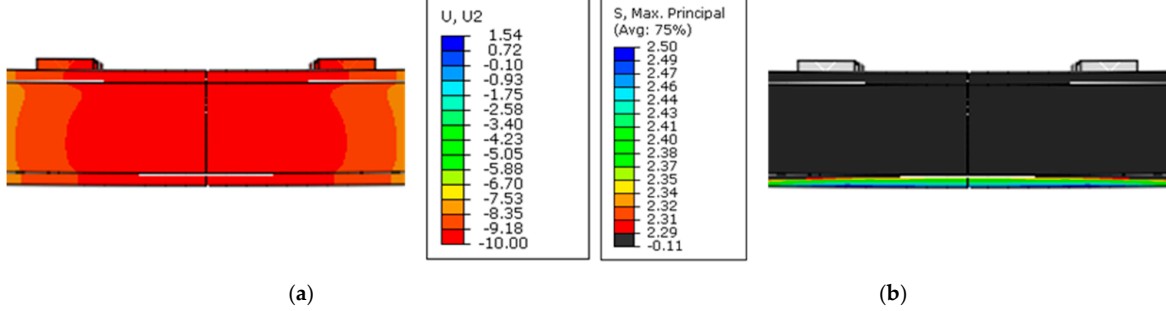

(**a**)　　　　　　　　　　　　　　　　　　　(**b**)

**Figure 9.** Start of the third stage at a load of 10 kN. (**a**) Vertical displacement (U, mm) in the middle of the beam; (**b**) the matrix cracking stress (S, in MPa) reaches through the entire cross-section of the tensile face in the constant moment area.

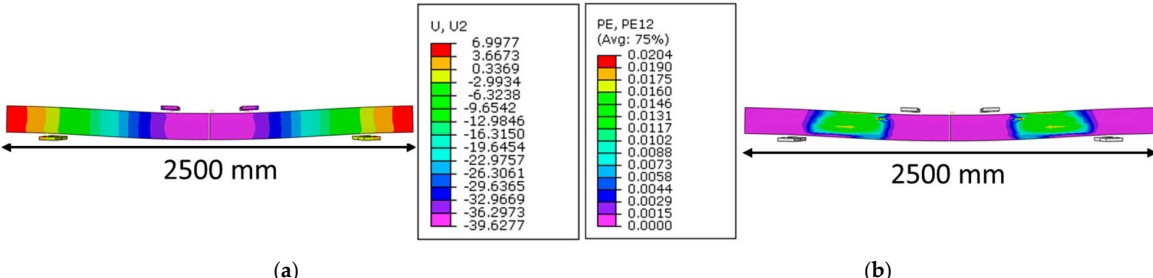

(**a**)　　　　　　　　　　　　　　　　　　　(**b**)

**Figure 10.** (**a**) Vertical displacement (U, in mm) at the start of the fourth stage (25.5 kN), and (**b**) plastic shear strain (PE [-]) of the core at a load of 25.5 kN.

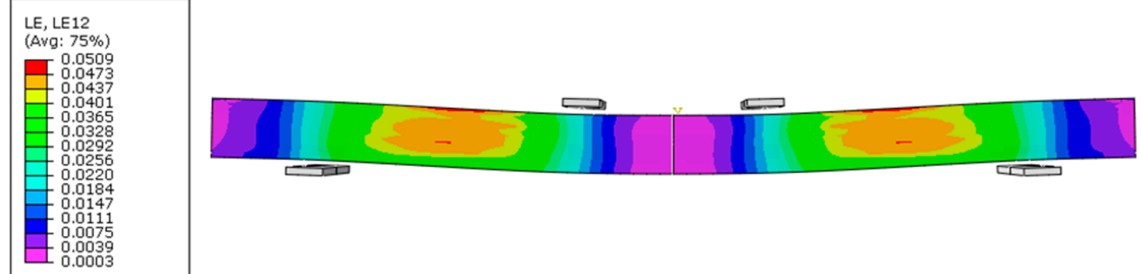

**Figure 11.** Total shear strain at a load of 25.5 kN.

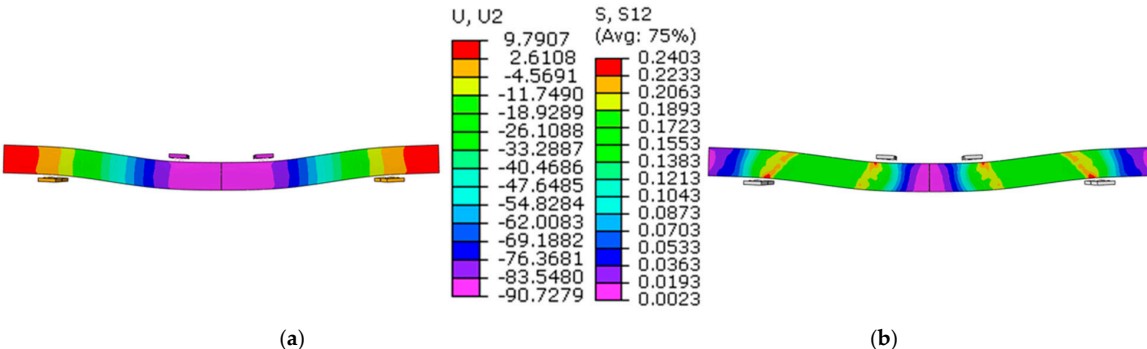

(**a**)　　　　　　　　　　　　　　　　　　　　　　　　　　　(**b**)

**Figure 12.** (**a**) Vertical displacement (U, in mm) of 91 mm, and (**b**) shear stress in the core at a displacement of 91 mm.

### 4.2. Experimental Results

A good correspondence between the numerical prediction and the experimental results was obtained; however, only three of the four stages were clearly visible in the experimental results.

In the first stage, both the faces and the core behaved linearly elastically. After reaching the matrix cracking stress in the area of the constant moment, the bottom face started to crack, indicating the start of stage II in Figure 7a. Figure 13 shows the longitudinal strains in the sandwich beams measured by both DIC systems, and therefore shows the appearance of the first crack after reaching the ultimate matrix cracking strength. These strain plots must be interpreted carefully. Strain results of the DIC technique were calculated from the average displacements, meaning that the displacements in the neighbourhood of cracks were responsible for apparent high strains at the location of the cracks. In reality, however, the strain in a crack is zero, so strain colormaps, as in Figure 13, can only be used to identify crack patterns; no significance should be attributed to the value of the strain in the vicinity of a crack. Simultaneously, the core showed a linear elastic shear strain, as shown in Figure 14.

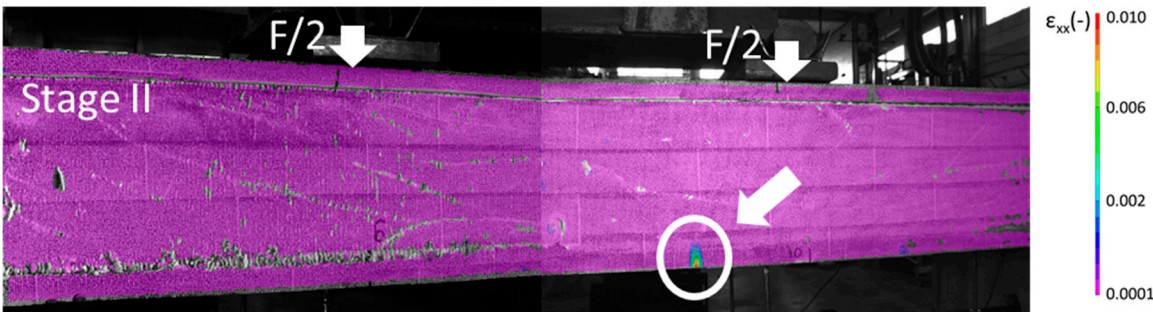

**Figure 13.** Longitudinal strain at the start of stage II, when the first crack appears at a bending load of 5 kN.

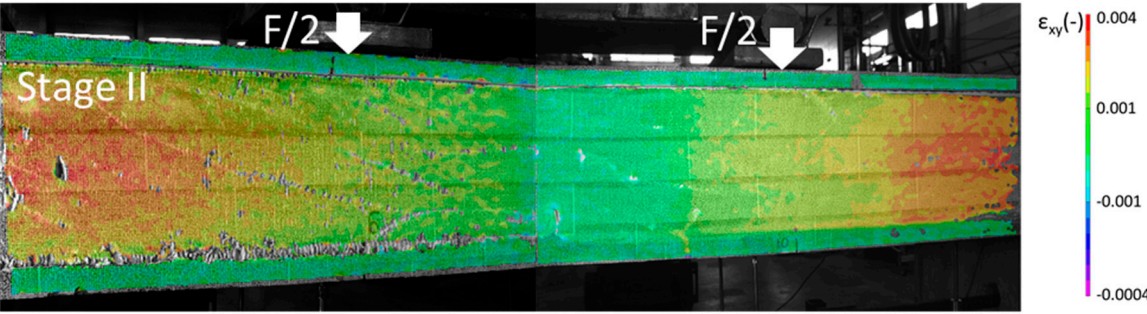

**Figure 14.** Shear strain of the core in stage II, at a bending load of 5 kN.

During stage III, cracking and propagation of the cracks occurred as shown in Figure 15. Since the highest tensile stress occurred in the area with the constant moment, most of the developed cracks are located between the loading beams.

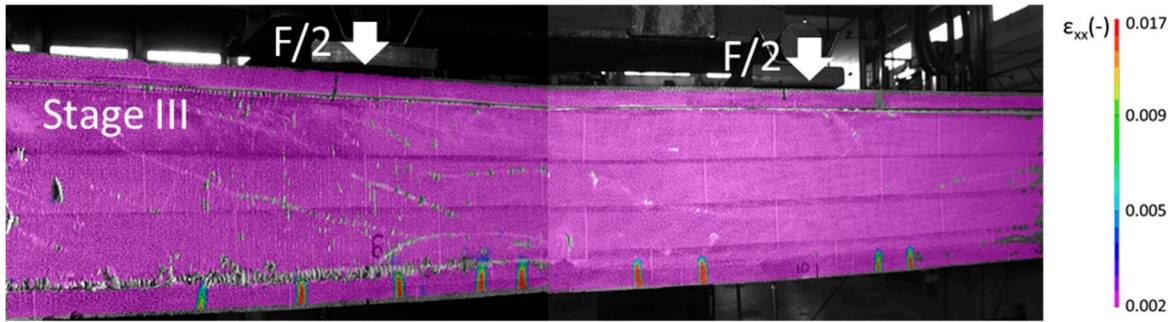

**Figure 15.** Longitudinal strain $\varepsilon_{xx}$ at a load of 16 kN.

At a load of 26 kN (stage IV), the tensile face showed multiple cracking and the saturation of cracks between the loading beams (see Section 2.1.1), while the core reached a shear strain of 0.019, which is equal to the plastic shear strain of XPS. Both the tensile strain of the tensile face and the shear strain of the core are shown in Figures 16 and 17. The numerical model, however, predicted the plastic shear deformation of the core at a load of 25.5 kN. The observed phenomenon, plastic shear deformation of the core, was the same for the experiments and the numerical prediction. Also, the degradation of the stiffness corresponded well. The core continued deforming plastically, until its ultimate shear stress was reached and failure of the core occurred.

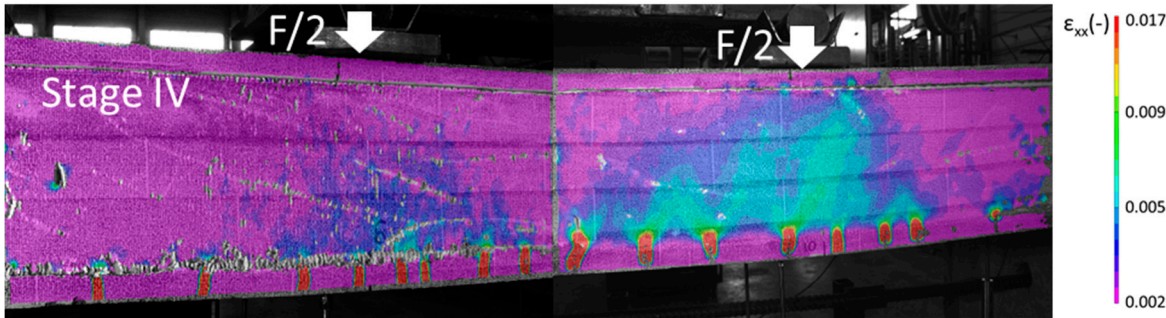

**Figure 16.** Longitudinal strain $\varepsilon_{xx}$ at a load of 26 kN.

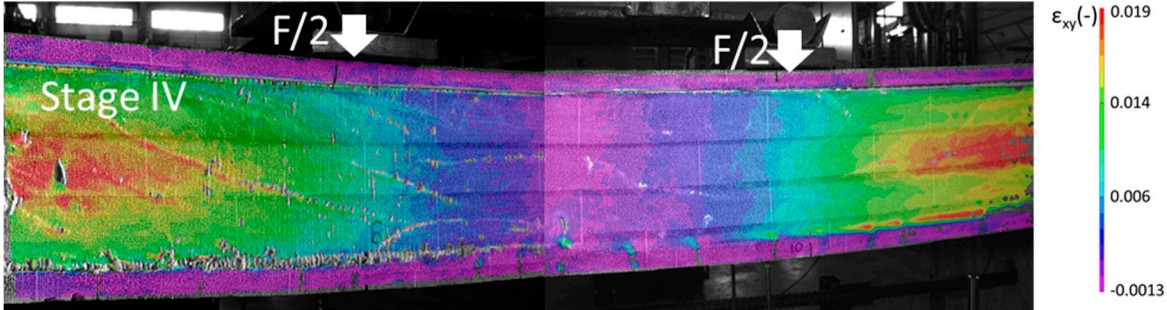

**Figure 17.** Shear strain $\varepsilon_{xy}$ at a load of 26 kN.

*4.3. Strain Comparison with the Numerical Model*

The numerical prediction of the bending behaviour of sandwich beams, indicated by the orange dotted line in Figure 7, was established as explained in Section 3. In terms of force-displacement behaviour, a good agreement between the experimental and numerical results was obtained. Due to the

full-surface DIC analysis, a more detailed comparison between the experimental results and numerical model could be performed in terms of strains, leading to a thorough validation of the model.

The strains in the TRC faces during the experiments were derived from the DIC results by artificially adding an extensometer between the loading beam and the middle of the beam (in the area of the constant moment). These artificial extensometers calculated the strain between two points, by dividing the measured displacements during loading by the initial calculated distance. In this way, an average shear strain in the area with the constant moment was obtained and quantified during the experiment. Since two DIC systems were used to monitor the full beam, the average of both systems was calculated, so that the complete area of the constant moment was covered (see Figure 18b). The strains of the numerical model were determined by calculating the ratio of the difference in longitudinal displacement between the middle of the beam and a point below the loading pin at the tensile face, and the initial distance between the same point (250 mm).

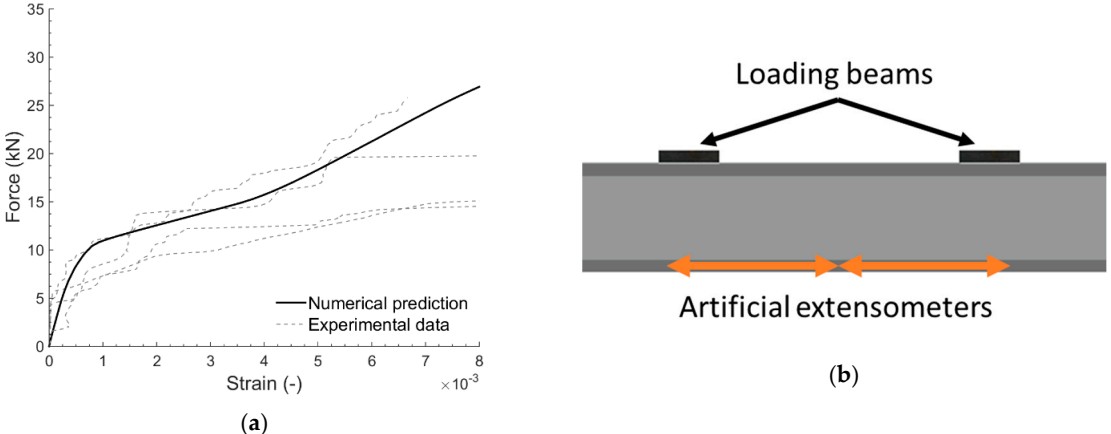

(**a**)
(**b**)

**Figure 18.** (**a**) Longitudinal strain in the tensile face of sandwich beam, obtained from the experimental results and numerical results; (**b**) schematic presentation of the artificial extensometers added on the DIC images.

The comparison of the experimental and numerical strains in the tensile TRC face are shown in Figure 18a. Both experimental and numerical longitudinal strains in the tensile face of the sandwich beam showed non-linear behaviour, as depicted in Figure 18a. The numerically implemented tri-linear tensile behaviour of the TRC is clearly visible in the tensile face of the sandwich beam in the bending of the numerical model. The number and place of the cracks in the tensile face cannot be predicted, which resulted in scattered experimental strain results in the tensile face. Due to this scatter, the experimental results showed a less pronounced tri-linear behaviour, but still follow the numerical tendency and showed the non-linear behaviour as predicted by the numerical model.

The core behaved plastically between the loading beams and the supports in the fourth stage, meaning that the plastic shear strain was reached at the beginning of this stage. Figure 19 shows the shear strain of the numerical prediction at a load of 26 kN, which was the start of the plastic shear behaviour in the experiments. These strains were compared with the experimental shear strains shown in Figure 19 and gave an identical strain distribution, with the maximal shear strain at the middle of the beam's height. Nonetheless, in the numerical analysis shear strain concentrations were noticed at the interface between the face and core, causing an overestimation of the shear strain. These concentrations were assumed to be due to the perfect bond, as simulated in the numerical model.

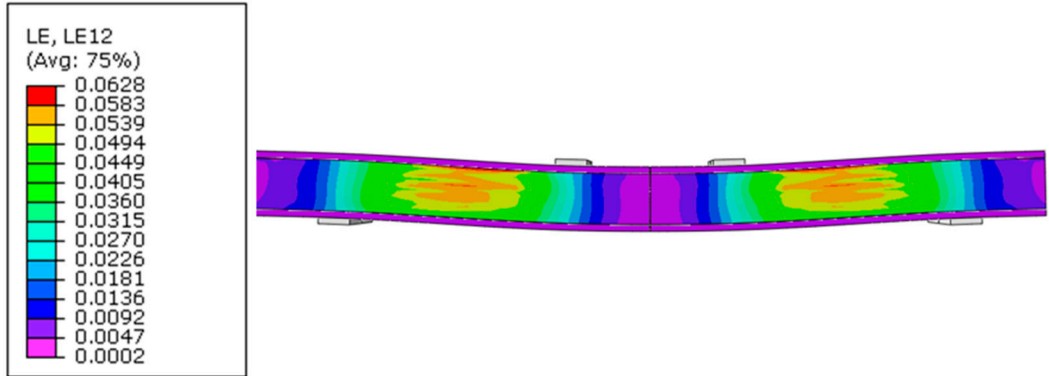

**Figure 19.** Shear strain in the core predicted by the numerical model, at a load of 26 kN.

Figure 20 shows the shear strains in the core at ultimate failure for both the experimental and numerical results of the sandwich beam. The same tendency was noticed for the numerical and experimental shear strain, which was limited in the area with the constant moment and increased outside the loading beams, where the highest shear forces were expected. Due to the perfect bond hypothesis in the numerical model, the shear strains were slightly overestimated.

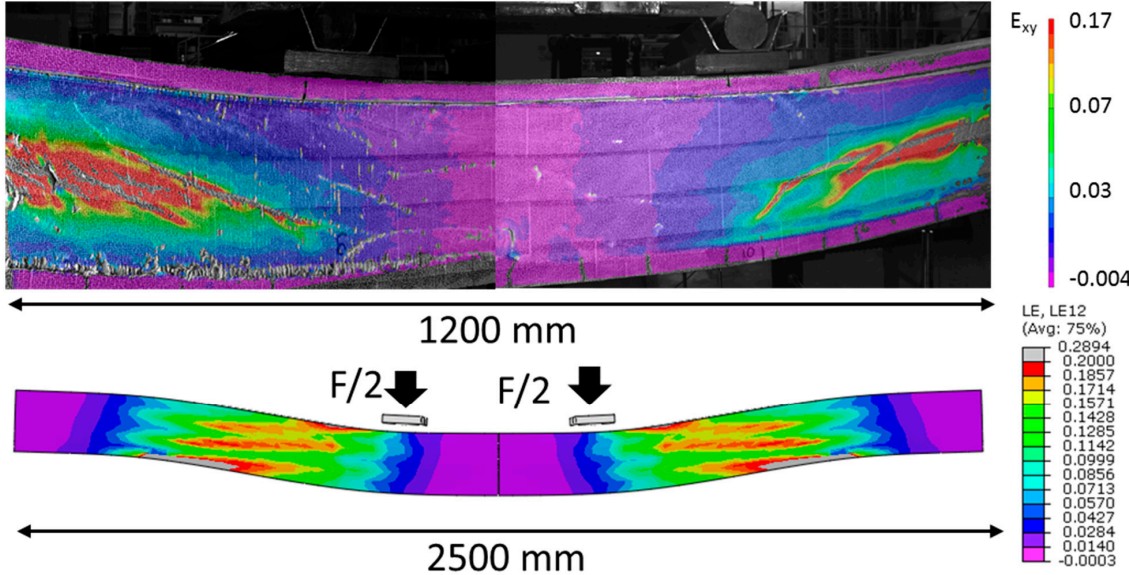

**Figure 20.** Comparison of the shear strain at failure load.

## 5. Conclusions

This paper presents a detailed comparison between a numerical prediction and experimental results of TRC sandwich beams under four-point bending by means of DIC. The numerical model considered the non-linear behaviour of both the TRC faces and the XPS foam core. A first comparison was made based on the force-displacement behaviour, which gave a good correspondence between the numerical prediction and the experimental results.

The stress and strain predictions of the numerical model identified multiple stages in the bending behaviour of the sandwich beams, which were confirmed by the experimental results. In the first stage, both the TRC faces and the XPS core behaved linearly elastically. Once the TRC tensile face started to crack, the second stage started. The four-point bending tests ended when the sandwich beams failed by shear failure in the core.

Thirdly, the tensile and shear strains obtained from the experiments and numerical simulation were compared. The TRC tensile strain was taken in the lowermost layer of the tensile face in the

area of the constant moment, both for the numerical model and for the experiments. Both the tensile strain in the face and the shear strain distribution in the core corresponded well, indicating that the numerical model can reliably predict the experimental strains.

In conclusion, this paper showed how the use of the digital image correlation measurement technique allows for a full-field displacement and strain measurement of TRC sandwich beams, as well as the monitoring of the evolution of the crack pattern in the TRC faces. This detailed validation of the established finite element model contributes to the state of the art on the behaviour of TRC sandwich panels.

**Author Contributions:** Conceptualization, J.V., T.T. and P.V.I.; methodology, J.V. and T.T.; formal analysis, J.V.; investigation, J.V.; writing—original draft preparation, J.V.; writing—review and editing, T.T., J.W., M.D.M.; M.E.K.; P.K.; visualization, J.V.; supervision, T.T., J.W.; project administration, T.T.; funding acquisition, D.V.H.

**Funding:** This research was funded by Agentschap voor Innovatie en Ondernemen (VLAIO) grant number IWT140070.

**Acknowledgments:** The authors gratefully acknowledge Agentschap voor Innovatie en Ondernemen (VLAIO) for funding the research, and the Belgian Building Research Institute (BBRI) for the collaboration in the tests performed in this research.

**Conflicts of Interest:** The authors declare no conflict of interest.

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
