# Peer review of "Validation of a Numerical Bending Model for Sandwich Beams with Textile-Reinforced Cement Faces by Means of Digital Image Correlation"

_applsci, doi:10.3390/app9061253_

Round 1

Reviewer 1 Report

The reviewer considers that this paper includes interesting experimental results and important findings. This paper also gives well the details of the experiments and analysis. The reviewer requests some revises.

1.    Figure 1 (b) and (c): Please show the dimensions and unit clearly.

2.    Figure 2 (a): Please show the texts clearly.

3.    Lines 88-89: “indicated by I, II and II” -> “indicated by I, II and III”

4.    Lines 93-94: Though the authors describe that “only three are shown”, there are many curves in the figure.

5.    Numerical model definition: The authors should explain the modeling of the boundaries between the different materials. Are the nodes shared by two materials elements?

6.    From Figure 8 to 11: Please indicate the units.

7.    There is no explanation about Figure 13, 14, and 15 in the body text.

Author Response

Dear reviewer,

Please find attached the rebuttal letter.

Best regards,

PhD student, Ir.Arch 

Jolien Vervloet

Reviewer 2 Report

Journal: Applied Sciences

Manuscript ID: applsci-448362

Title: Validation of a numerical bending model for sandwich beams with Textile Reinforced Cement faces by means of Digital Image Correlation

The topic addressed in the paper entitled “Validation of a numerical bending model for sandwich beams with Textile Reinforced Cement faces by means of Digital Image Correlation” falls within the technical area covered by Applied Sciences. In my opinion, the manuscript will be worthy of publication after the requested major improvements. In the attached document, the authors will find my detailed comments.

Author Response

(The authors gave the same response as above.)

Reviewer 3 Report

The submitted article with ID “applsci-448362” and title “Validation of a numerical bending model for sandwich beams with Textile Reinforced Cement faces by means of Digital Image Correlation” contains original experimental and numerical contributions to the study of sandwich panels with Textile Reinforced self-compacting Cement (TRC) faces under flexural loading. The experimental procedure includes interesting measurements of strains and deformations of the tested panels using a monitoring Digital Image Correlation (DIC) system. Comparisons between these experimental measurements and the strains derived from the performed numerical analyses are also presented. The current study deals with a topic that is very interesting for the researchers and derives to useful concluding remarks for the practicing engineers concerning the application and the structural capacities of lightweight insulating sandwich panels. The experimental program is well-planned and the numerical analysis is well-presented. The manuscript is also well-structured and easily understood. As an overall and based on the aforementioned comments the paper is worthy of publishing. The following suggestions and clarifications would improve the article:

1.    The faces of the examined sandwich panels were made of TRC plates consisting of a self-compacting ordinary Portland cement matrix cast onto the fiber textiles. The advantages of the use of self-compacting mortar (instead of a common cementitious mortar with ordinary compacting properties) in these panels should further be highlighted. Recently, applications of self-compacting concrete, mortar and cement mixtures have been extended for the rehabilitation of damaged or deficient reinforced concrete structural members using thin jackets made of self-compacting mortars with light reinforcement. Introduction could be slightly enriched with some comments concerning these issues and applications (some relative references that could be considered are also suggested below). This way, the subsequent impact of this study on the state of the practice along with the research significance and the objectives of the paper would further promoted. Further, the Authors could consider including the term of “self-compacting” in the title of their manuscript.

2.    Most figures of the paper are precise and very helpful. It would be useful Figure 5 to present the cross-section of the panels along with some details concerning the dimensions, materials, etc. A supplementary photograph of the test rig with the DIC system would also be greatly appreciated.

3.    Figure 7 could be enriched with a couple of additional photographs demonstrating the first crack initiation and the development of multiple cracks in the tensile face of the beam (end of stage I) and some of the next loading - damage stages (stages III and IV for example). These photographs could be numbered, noted in the force versus displacement diagram of Figure 7 and commented in the manuscript. It is stressed that although there are some comparative “photographs” in Figure 12-15 from the DIC processing, the additional photographs in Figure 7 would be different. The cracking and damage propagation of the tested panel is of great interest since these tests are also compared successfully with the numerical simulation.

4.    The sole equation (3.2.4) of the manuscript is rather trivial and therefore it could be included and described in the text.

5.    The purpose and the configuration of the extensometers added on the DIC images should be clarified and further discussed.

Suggested supplementary bibliography based on #1 comment (order by date):

-   “Rehabilitation of shear-damaged reinforced concrete beams using self-compacting concrete jacketing”, ISRN Civil Engineering, 2012.

-  “Application of a reinforced self-compacting concrete jacket in damaged reinforced concrete beams under monotonic and repeated loading”, J. Engineering, 2013.

-  “Behaviour of rehabilitated RC beams with self-compacting concrete jacketing – Analytical model and test results”, Construction and Building Materials, 2014.

-   “Experimental study of the effectiveness of retrofitting RC cylindrical columns using self-compacting concrete jackets”, Construction and Building Materials, 2016.

-  “Flexural strengthening of damaged RC T-beams using self-compacting concrete jacketing under different sustaining load”, Construction and Building Materials, 2018.

-  “Exploitation of Ultrahigh-Performance Fibre-Reinforced Concrete for the Strengthening of Concrete Structural Members”, Advances in Civil Engineering, 2018.

Author Response

(The authors gave the same response as above.)

Reviewer 4 Report

The topic is of potential interest for journal readers. Authors should clarify the novelty of their paper, since they refer totally to previous experimental tests and numerical FEM modelling, already published; so it is not clear the novelty (is it the DIC analysis?). The discussion of DIC is limited to few pages; furthermore I suggest to improve the references to DIC and information on code used and settings, for instance adding some relevant papers, among others on application of DIC to FRC: doi: 10.1016/j.compositesb.2017.05.075 .

Line 76. Please clarify the evaluation of critical fibre volume fraction.

Line 88. Please check, Stage III is missing and stage II is repeated twice.

Line 92. Please define E-modulus; the transition strain is missing between stages II and III.

Line 161. Please clarify why Poisson ratio is equal to 0.5.

Line 175. Please clarify the choice of a time period of 10 and a mass scaling factor of 0.000001.

Figure 7a, I suggest to plot (second scale of x axis) also in terms of displacement over length of beam between supports.

Line 251. Please revise “validation of the numerical which was still missing…”

Author Response

(The authors gave the same response as above.)

Round 2

Reviewer 1 Report

The items which were pointed out by the reviewer are sufficiently improved.

Author Response

We would like to thank the reviewer again for the useful comments. 

Best regards,

Ir. Arch. Jolien Vervloet

PhD student at the department of Mechanics of Constructions and Building materials

Reviewer 2 Report

I have read with interest the authors’ responses to my comments.

However, I am still not satisfied with the two responses:

Comment 10

In this comment I was expected not only to specify the size of the mesh, but also to provide information about its calibration.

Comment 11

I am still not sure if the agreement between the numerical and experimental shear strains can be considered as good. I find quite surprised that the maximum strains in the numerical model are located near the contact layer between the XPS core and top TRC layer, by contrast to the experimental results. The maximum strains of DIC occur in the mid-height of the core in the shear span and are several times higher than the strains in the numerical model. Also in the area with constant moment, some differences in the strain distribution are clearly visible. More information about the results agreement would be provided by the comparisons of strains along the beam length, at mid-height.

I believe that the model of the XPS core as well as the contact model between the sandwich panels are not really reproducing well the experimental behaviour. Nevertheless, please try to explain the numerical results obtained.

Author Response

We would like to thank the reviewer again for the useful comments.

Best regards,

Ir. Arch Jolien Vervloet

Department of Mechanics of Materials and Constructions, Vrije Universiteit Brussel

Reviewer 3 Report

As it was stated in the first review, the paper is very interesting, well-structured, easily understood and derives to useful concluding remarks for the practicing engineers. The experimental program is well-planned and the numerical analysis is well-presented. Further, the revised paper (applsci-448362-peer-review-v2) has extensively been improved and enriched. New comments, paragraphs, Refs and Figures have been added. The efforts performed by the Authors to consider all the recommendations and to respond to all the criticisms of the previous reviews are greatly appreciated. Hence, acceptance of the paper for publication in the journal without further re-review is recommended.

Author Response

(The authors gave the same response as above.)

Reviewer 4 Report

Authors solved my concerns

Author Response

(The authors gave the same response as above.)

Round 3

Reviewer 2 Report

I believe that the manuscript has been significantly improved and now warrants publication in Applied Sciences.